# The Relationship between Functional Movement Quality and Speed, Agility, and Jump Performance in Elite Female Youth Football Players

**DOI:** 10.3390/sports12080214

**Published:** 2024-08-06

**Authors:** Dan Iulian Alexe, Denis Čaušević, Nedim Čović, Babina Rani, Dragoș Ioan Tohănean, Ensar Abazović, Edi Setiawan, Cristina Ioana Alexe

**Affiliations:** 1Department of Physical and Occupational Therapy, “Vasile Alecsandri” University of Bacău, 600115 Bacău, Romania; alexedaniulian@ub.ro; 2Faculty of Sport and Physical Education, University of Sarajevo, 71000 Sarajevo, Bosnia and Herzegovina; denis.causevic@fasto.unsa.ba (D.Č.); nedim.covic@fasto.unsa.ba (N.Č.); ensar.abazovic@fasto.unsa.ba (E.A.); 3Department of Physical Rehabilitation & Medicine (Physiotherapy), Post Graduate Institute of Medical Education and Research, Chandigarh 160012, India; says2babina@gmail.com; 4Department of Motric Performance, Transilvania University of Brașov, 500036 Brașov, Romania; 5Faculty of Teacher Training and Education, Suryakancana University, Cianjur Kidul 43216, Indonesia; edi_setiawan@unsur.ac.id; 6Department of Physical Education and Sports Performance, “Vasile Alecsandri” University of Bacău, 600115 Bacău, Romania

**Keywords:** FMS, mobility, stability, athletic performance, female football, soccer

## Abstract

The association between movement screening and physical fitness testing in athletes is conflicting, and therefore, this study aimed to examine the relationship between Functional Movement Screen (FMS) performance and physical performance in elite female youth football players. Twenty-two players from the national U16 team of Bosnia and Herzegovina underwent FMS and physical performance tests, including speed, agility, and jump assessments. Jump and speed performance score correlated well with ASLR, while the overall FMS score was not associated with any of the performance variables. These findings suggest that while certain movement patterns may impact athletic performance, the relationship between movement screening and physical performance is delicate. Coaches and practitioners should consider individual variations and sport-specific demands when interpreting FMS results in order to optimize and maximize athlete performance and reduce injury risks.

## 1. Introduction

Football is an immensely popular and an intense sport that involves high-demand tasks such as landing, cutting, and rapid change of direction. Players are also expected to have higher performance in strength, speed, endurance, and power [1]. This necessitates a holistic approach to training and assessment.

The screening tests that can detect modifiable intrinsic risk factors for musculoskeletal injury have always been of interest to the practitioners of sport and exercise medicine. It has been documented previously that football players’ performance is affected by their ability to execute movements correctly, which is emphasized in the literature as crucial for long-term safety and effectiveness [2,3].

The variety of tests used to evaluate different sport-specific parameters do not directly implicate any information on the efficient movement proficiency of the player. For resolving this, a battery of assessment tools was developed with Functional Movement Screening (FMS). Pre-participation screening and sports performance can be linked together with the use of FMS, the most popular movement screening technique used by specialists to evaluate an individual’s fundamental movement patterns [4]. By assessing any asymmetries, the dysfunctional patterns and higher musculoskeletal injury risks can be identified with FMS. A score of <14 on FMS has been linked to greater chances of injury [5,6]. However, this cutoff score can be influenced by multiple factors, like gender, age, body composition, type of sport, training state, and diet, among others. Through such stratification, targeted interventions can be delivered to groups considered “at-risk” in order to reduce the likelihood of injury, thereby enhancing the performance.

FMS has gained popularity as a comprehensive assessment tool in sports science [4]. However, its relationship to sport-specific performance metrics remains a subject of debate among researchers and practitioners [7,8]. While FMS aims to evaluate fundamental movement patterns, it is essential to consider how these patterns translate into actual athletic performance. The correlation between FMS scores and various sport-specific parameters, such as speed, agility, and power, has yielded inconsistent results across different studies and sports disciplines [9,10]. This variability in findings highlights the need for sport-specific investigations to determine the true value of FMS in predicting and enhancing athletic performance.

The relationship of movement screening with physical fitness testing in athletes is conflicting in the available literature for different sports. Zhang et al. [11] have reported the inaccuracy of FMS in predicting jump and sprint performance, while few studies found FMS to correlate significantly with vertical jump [10], agility [2,10], and lower limb strength [9]. However, such an association in young female football players, per se, is so far under-explored.

This study is therefore an effort to provide a clear picture of the strength and conditioning coaches and the treating physical therapists of whether any association exists between physical performance and efficient movement (FMS) in young female football players. Therefore, the present study aimed to investigate the correlations between functional movement quality (FMS) and speed, agility, and jump performance in elite female youth football players.

## 2. Materials and Methods

### 2.1. Experimental Approach to the Problem

This was a transversal observational and descriptive study. This study obtained pre-season measures of physical performance (speed, planned agility, and vertical jump) and movement quality (FMS) in elite junior football athletes in Bosnia and Herzegovina. A cross-sectional analysis was conducted on each participant to examine the relationship between their movement quality and physical performance metrics. To address the research questions, this study employed a multivariate correlation analysis approach.

### 2.2. Participants

Twenty-two elite female youth football players (age: 15.59 ± 0.57) from the national U16 team of Bosnia and Herzegovina participated in this study. All participants had a minimum of 5 years training experience. Before this study, all athletes were provided information about the procedures and study objectives. Subjects with a history of recent (previous 30 days) injury or surgery prohibiting full participation in the regular training schedule were excluded. Participants were excluded if they were in their menstrual cycle. The study was approved by the Ethics Committee of the Faculty of Sports and Physical Education, University of Sarajevo (No: 01−2603/22; 1 July 2022) and was carried out in accordance with the Declaration of Helsinki. After the procedures were fully explained, parents provided written consent for their minor children to participate in the study, ensuring complete ethical compliance and explicitly stating that participants could withdraw from the study at any time without consequence.

### 2.3. Measurement Procedure

All the tests/measurements were performed at the Institute of Sport at the Faculty of Sports and Physical Education, University of Sarajevo. For this study, an appropriate training/testing cycle was developed, consisting of a single day of testing. The testing session was performed on the last day of 5 days of national training camp in the morning between 09:00 and 12:00. The research procedure started with the evaluation of body composition (body height, body mass, percentage of body fat (PBF), body mass index (BMI), and fat free mass (FFM)) by digital stadiometer (InBody BSM 370; Biospace Co., Ltd., Seoul, Republic of Korea) [12], and a direct segmental high-frequency bioelectrical impedance scale (InBody 720; Biospace Co., Ltd., Seoul, Republic of Korea). A general independent warm-up was provided by the national selection strength and conditioning coach. After warming up, subjects completed the FMS movement assessment and a battery of physical performance tests. The physical performance tests included a 5, 10, 20, and 30 m sprint, countermovement jump—free arms (CMJ free arms) height, zig-zag agility test, and *t*-test. All physical performance testing was conducted by trained staff with previous experience in physical performance assessment.

### 2.4. Functional Movement Screen (FMS)

Participants were screened using the functional movement screen protocol (FMS) that used 7 movement patterns [4,13]. FMS reliability has been previously established [14]. Each participant performed three trials of every movement pattern. An experienced rater, with two years of screening experience, evaluated these trials in real time using a 4-point scale (Table 1), adhering to the FMS rater manual and established research protocols [4]. The testing procedure incorporated rest periods of approximately 5 s between trials, 1 min between different tests, and participants returned to the starting position after each trial. In line with FMS criteria, the highest score from the three trials was recorded for subsequent analysis. For bilateral tests, the lower of the two scores was documented.

### 2.5. Speed

The assessment of running speed involved participants performing 30 m maximal sprints. Each sprint began from a stationary standing position, with the athlete’s lead foot positioned 20 cm behind the initial photocell. Four photocells (Witty, Microgate, Bolzano, Italy) were used to measure 5 m, 10 m, 20 m, and 30 m maximal sprint times (in seconds) [15,16]. Each player performed two trials (ICC > 0.81) with a 3 min rest in between, and the best of the two was analyzed. Photocells were placed at a height of 120 cm [17].

### 2.6. Agility

Agility was evaluated using the same photocells (Microgate, Bolzano, Italy) through the zig-zag and *t*-test, as recommended by previous studies [18,19,20].

The zig-zag agility test was selected for its ability to assess multiple facets of agility, including short acceleration, deceleration, and balance control. Conducted on an outdoor football field, the test comprised four 5 m sections totaling 20 m of linear sprinting. Cones were positioned at 100° angles to mark the course, following established protocols from previous research [20]. Each participant completed the test twice (ICC > 0.84), with the best score being recorded for analysis.

The T-test assessed athletes’ agility through a series of movements. Participants began with their lead foot 20 cm behind the initial photocell, then sprinted 9.14 m forward to touch a center cone with their right hand. They then shuffled 4.57 m left to touch a second cone, followed by a 9.50 m right shuffle to a third cone, before shuffling back 4.75 m to touch the center cone with their left hand. Finally, they ran backward to the starting point [20]. The timing was recorded in seconds using photocells (Microgate, Bolzano, Italy) at the starting line. Each athlete completed two trials (ICC > 0.79) with a 3 min rest between attempts, and the best time was recorded.

### 2.7. Jumping Performance

For assessment of the jumping performance, the countermovement jump with free arms (CMJ free arms), which was found to be reliable previously [21], was performed [15]. Protocols included two data collection trials and were performed using Optojump Next system (Microgate, Bolzano, Italy). At the beginning, players started from an upright standing position with feet shoulder-width apart and hands next to the body for arm swing. They performed a preliminary downward countermovement to a self-selected depth by flexing the hips and knees. Players then immediately extended their hips and knees to execute a vertical jump. After the jump, the player returned to the starting position. This procedure was repeated twice (ICC > 0.91), and the best of the two attempts was used for analysis. The trial was considered invalid if knee flexion occurred upon landing.

### 2.8. Statistics

The results were presented as mean ± standard deviation (SD). Spearman’s rank correlation analysis was used to quantify associations between FMS—tests and sprint, agility, and jump performance variables. The strength of correlation was determined as small (<0.29), moderate (0.3–0.49), large (0.5–0.69), or very large (>0.7) [22]. The statistical analysis was performed using SPSS (Version 22. For Windows, Armonk, NY, USA. IBM Corp.). Statistical significance was set at *p* < 0.05.

## 3. Results

Twenty-two players (four strikers, nine midfielders, seven fullbacks, and two goalkeepers) completed the testing procedure and were included in a final analysis. Table 2 shows the descriptive statistics for the morphological, performance, and FMS variables.

Table 3 shows all the correlation coefficients between morphological, performance, and FMS variables. Significant correlation was found between shoulder mobility and two agility (−0.39 and −0.42) and two sprint (−0.42 and −0.38) tests and variables, respectively. Furthermore, the active straight leg raise test correlated significantly to jump (0.45) and speed (ranging −0.41 to −0.47) performance and the in-line lounge test correlated significantly to the zig-zag test (−0.44). It is interesting to note that no correlation was found between the overall FMS score and any performance variable.

## 4. Discussion

This study aimed to elucidate whether a relationship exists between physical performance and efficient movement (FMS) in young female football players. Results suggested that the FMS score, surpassing the cutoff of 14.5 as proposed by Zhang et al. [11], was sufficient to indicate good sensitivity and a lower false-positive rate of FMS in female football players.

Overall, the FMS score had a very weak positive influence on zig-zag and T-test agility and 5 m, 10 m, 20 m, and 30 m sprint times along with CMJ free arms performance. These findings contrast with those of Zhang et al. [11] for elite young female players, who reported a moderate correlation between FMS score and 10 m and 20 m sprint times. Prior studies examining this association have yielded ambiguous outcomes. Several investigations have identified slight to moderate correlations between FMS scores and diverse physical performance metrics, such as jumping, agility, or speed assessments, specifically in female populations [2,8,11,23,24]. The inconclusive results could likely be attributed to the participants’ level of maturity, as it is widely acknowledged that rapid growth during adolescence can impact performance in young athletes. Despite the lack of significant difference in the total FMS score between playing position groups, this factor could still provide a plausible explanation for the variability in results. Moreover, in our study, participants exhibited notably higher body size (21 vs. 19 kg/m^2^) and had better sprint and FMS scores but lower jump performance when compared to the participants in a previous study [11]. This difference in performance could potentially be attributed to higher muscle mass among our study participants, who were approximately 9 months older and had more training experience. Our findings suggest that in young athletes, functional movements may play a more significant role in performance, whereas in more mature players, performance may rely more on power and strength. Moreover, Bennett et al. [25] demonstrated the potential of FMS during athletic performance optimization since FMS can determine acceptable movement capabilities. They found a significant, albeit slight, association of FMS score to the 5 m sprint time as well as the agility performance. Additionally, they observed a significant increase in both FMS score and speed performance after one year, although no significant relationships were found between these improvements, supporting the importance of our assumption previously explained. These findings underscore the significance of adequate movement ability to enhance physical performance, but it should not be underestimated since this potentially can reduce the risk of injuries [26]. As stated by Bennet et al. [25] “If movement quality does play a role in physical performance capacity, it is likely to be only one small part of the puzzle”.

Observing relations between each test, it was evident that a higher in-line lunge score was associated with better zig-zag results, higher shoulder mobility with better agility (zig-zag and T-test), and acceleration capabilities (5 and 10 m), with small but significant correlations, respectfully. These results align with those presented by Zhang et al. [11]. Higher active straight leg raise score contributed to better CMJ and maximal speed (10 m, 20 m, and 30 m) test performance. As far as athletic performance goes, movement quality has a limited impact. Although the FMS evaluates specific isolated fundamental movement patterns, it cannot predict abilities like agility and execution speed [27]. Moreover, the precise technical skill of force application significantly impacts sprint performance, highlighting the inadequacy of fundamental movement alone.

Moreover, the present study further demonstrated that increased range of motion (ROM) in shoulders and dorsal lower body did correlate with enhanced agility and acceleration physical performance. Such findings could be the result of increased ROM ability to facilitate a more extensive arm swing during sprinting, leading to increased propulsion and higher speed, and enabling better coordination between upper and lower body movements, enhancing the transfer of force and thus improving power production. Additionally, increased shoulder mobility contributes to better balance and stability, crucial for maintaining proper body position [28]. Optimal shoulder and lower limb ROM are essential for proper body mechanics and posture, which can enhance movement efficiency. However, individual responses may vary, highlighting the importance of tailored training programs focusing on flexibility, mobility, and strength to maximize the benefits of increased shoulder ROM on athletic performance. Consequently, strength and conditioning coaches should recognize that ROM assessments may offer good insights into the possible issues regarding low basic performance of female athletes [29].

As observed in a study by Alicea-Kulian et al. [30] using stepwise multiple regression analysis, deep squats predicted vertical jump height (in centimeters) most effectively (r = 0.416). Our results indicate that active straight leg raises correlate to vertical jump performance. These findings emphasize the significance of hamstring flexibility and hip functionality as potential predictors of performance variables, especially for power movements such as the vertical jump.

These results suggest that there are associations between FMS scores and certain physical performance measures in elite female youth football players, indicating the importance of functional movement quality in athletic performance.

The investigations of Functional Movement Screening (FMS) and its relationship with physical performance in young female football players have several potential reasons for variations in correlations between FMS and tests of strength, speed, and agility. Objective limitations arise when comparing outcomes across different studies in this population.

While football is played similarly worldwide, variations in training approaches and regimes can influence the development of diverse movement patterns and aspects of strength, speed, and agility. Moreover, identical, or similar movements may be cultivated using different training methods, impacting test results. Even slight differences in testing methodologies can lead to varying outcomes and levels of correlation. Our results align well with a series of studies that focused on the interrelationship between the components of agility, strength, explosive power, reaction time, and balance which highlighted that sports performance in team sports is conditioned by the level of development of all components of motor capacity [22,31].

Furthermore, young athletes with differing levels of experience may demonstrate varying correlation levels between functional movement tests and assessments of strength, speed, and agility. Those with greater experience may exhibit superior movement techniques, resulting in heightened correlations between tests. Genetic factors, body morphology, and anthropometric measures can also influence the levels of strength, speed, and agility, consequently affecting correlations with functional movement tests.

The current physical condition of players, including factors such as injuries or fatigue, can also significantly influence test outcomes and correlations between tests. Variations in measurement error or uncertainty can further impact correlations between tests. Therefore, for future studies, it is imperative to clearly state measurement errors to allow for comparison and interpretation of results.

This study has a few limitations. The study’s conclusions and statistical strength may be constrained by its limited participant pool of only 22 subjects. Larger sample sizes in future studies could provide more generalizable results. Additionally, the study employed a single day of testing, which might not capture the variability in performance that could occur over multiple testing sessions. Furthermore, the young athletes with differing levels of training experience may demonstrate varying correlation levels between functional movement tests and assessments of strength, speed, and agility. Those with greater experience might exhibit superior movement techniques, leading to heightened correlations between tests.

## 5. Conclusions

The findings of this study underline the importance and significance of a holistic approach. Furthermore, FMS assessment can be a valuable tool in injury prevention and its results do not influence physical performance since the correlation coefficients we found were, at most, moderate in elite female youth football players. Additionally, individual variations and factors such as training background, fitness level, and sport-specific demands should always be considered. A clear limitation of this study arose, namely, in the population studied. Considering that this is the national team, which has already undergone a detailed selection process, future studies should focus on lower-league female footballers.

## Figures and Tables

**Table 1 sports-12-00214-t001:** FMS criteria.

Score	Criteria
0	The participant experiences pain in any part of the body at any point during the test.
1	The participant either fails to finish the movement pattern or is unable to acquire the position required to conduct the movement.
2	The participant may finish the exercise but does so with compensation(s).
3	The participant can complete the activity accurately without any assistance.

**Table 2 sports-12-00214-t002:** Descriptive statistics and post hoc comparisons for body composition and physical fitness tests.

Variables	Total (*n* = 22)		Striker (*n* = 4)		Midfielder (*n* = 9)		Fullback (*n* = 7)		Goalkeeper (*n* = 2)	
	Mean	SD	Mean	SD	Mean	SD	Mean	SD	Mean	SD
Age (years)	15.59	0.57	15.90	0.27	15.42	0.66	15.55	0.51	15.90	0.84
Body height (cm)	168.30	6.43	167.05	9.98	167.64	6.79	169.24	4.30	170.45	7.99
Body mass (kg)	59.83	7.25	60.52	12.02	57.46	7.00	62.17	3.26	60.90	11.03
BMI (kg/m^2^)	21.06	1.57	21.52	2.14	20.38	1.46	21.71	1.26	20.90	1.83
PBF (%)	20.49	3.49	21.00	4.41	19.15	2.98	21.85	3.14	20.75	6.15
FFM (kg)	47.42	4.87	47.62	8.30	46.38	5.42	48.51	1.45	47.90	4.94
FMS score	16.50	2.22	17.25	1.50	15.88	3.05	17.28	1.11	15.00	0.00
Deep squat	2.45	0.59	2.50	0.58	2.33	0.70	2.71	0.48	2.00	0.00
Hurdle step	2.36	0.66	2.75	0.50	2.33	0.70	2.28	0.75	2.00	0.00
In-line lunge	2.59	0.59	3.00	0.00	2.44	0.72	2.71	0.48	2.00	0.00
Shoulder mobility	2.59	0.66	2.50	1.00	2.33	0.70	2.85	0.37	3.00	0.00
ASLR	2.59	0.50	2.25	0.50	2.55	0.52	2.71	0.48	3.00	0.00
TSPU	1.86	0.83	2.00	0.52	1.88	0.93	2.00	0.81	1.00	0.00
Rotary stability	2.04	0.21	2.25	0.50	2.00	0.00	2.00	0.00	2.00	0.00
CMJ free arms (cm)	28.88	4.01	28.47	2.80	29.95	3.86	27.70	5.19	29.00	3.39
Zig-zag (s)	6.29	0.29	6.41	0.26	6.24	0.24	6.22	0.39	6.49	0.91
T-test (s)	11.37	0.54	11.91	0.54	11.12	0.38	11.31	0.56	11.61	0.53
5 m sprint (s)	1.12	0.06	1.17	0.09	1.12	0.06	1.08	0.05	1.13	0.14
10 m sprint (s)	1.94	0.08	2.00	0.10	1.95	0.07	1.90	0.09	1.93	0.14
20 m sprint (s)	3.38	0.15	3.48	0.16	3.40	0.13	3.31	0.16	3.36	0.03
30 m sprint (s)	4.78	0.22	4.92	0.23	4.80	0.20	4.67	0.24	4.79	0.04

BMI: Body mass index; PBF: Percentage of body fat; FFM: Fat-free mass; FMS: Functional movement screen; ASLR: Active straight leg raise; TSPU: Trunk stability push-up; CMJ: Countermovement jump.

**Table 3 sports-12-00214-t003:** Correlation between FMS tests and sprint, agility, and jump performance variables.

Variables	CMJ Free Arms	Zig-Zag	T-Test	5 m	10 m	20 m	30 m
FMS score	0.17 (−0.27, 0.55)	−0.13 (−0.52, 0.31)	−0.18 (−0.56, 0.26)	−0.27 (−0.62, 0.17)	−0.14 (−0.53, 0.30)	−0.18 (−0.56, 0.26)	−0.21 (−0.58, 0.23)
Deep squat	0.22 (−0.22, 0.59)	−0.17 (−0.55, 0.27)	−0.15 (−0.54, 0.29)	−0.11 (−0.51, 0.33)	0.17 (−0.27, 0.55)	−0.17 (−0.55, 0.27)	0.09 (−0.34, 0.49)
Hurdle step	0.13 (−0.31, 0.52)	−0.15 (−0.54, 0.29)	−0.14 (−0.53, 0.29)	−0.19 (−0.57, 0.25)	−0.08 (−0.49, 0.35)	−0.11 (−0.51, 0.33)	−0.17 (−0.55, 0.27)
In-line lunge	0.28 (−0.16, 0.62)	−0.44 * (−0.72, −0.02)	−0.19 (−0.57, 0.25)	−0.10 (−0.50, 0.34)	−0.06 (−0.47, 0.37)	−0.10 (−0.50, 0.34)	−0.09 (−0.49, 0.34)
Shoulder mobility	0.20 (−0.24, 0.57)	−0.39 * (−0.70, 0.03)	−0.39 * (−0.70, 0.04)	−0.42 * (−0.72, 0.00)	−0.38 * (−0.69,0.05)	−0.29 (−0.63, 0.15)	−0.28 (−0.63, 0.16)
ASLR	0.45 * (0.03, 0.73)	−0.30 (−0.64, 0.14)	−0.30 (−0.64, 0.14)	−0.35 (−0.67, 0.08)	−0.41 * (−0.71, 0.01)	−0.47 * (−0.74, −0.06)	−0.46 * (−0.73, −0.05)
TSPU	0.06 (−0.38, 0.47)	−0.24 (−0.60, 0.20)	−0.17 (−0.55, 0.27)	−0.22 (−0.59, 0.22)	−0.16 (−0.54, 0.28)	−0.05 (−0.46, 0.38)	−0.07 (−0.48, 0.36)
Rotary stability	0.25 (−0.19, 0.61)	−0.02 (−0.44, 0.40)	−0.13 (−0.52, 0.31)	−0.18 (−0.56, 0.26)	−0.27 (−0.62, 0.17)	−0.23 (−0.59, 0.21)	−0.25 (−0.61, 0.19)

FMS: Functional movement screen; ASLR: Active straight leg raise; TSPU: Trunk stability push-up; CMJ: Countermovement jump. * Correlation is significant at the *p* < 0.05 level.

## Data Availability

The data presented in this study are available upon reasonable request from the corresponding author.

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
