# Peer review of "The Relationship between Functional Movement Quality and Speed, Agility, and Jump Performance in Elite Female Youth Football Players"

_sports, 2024, doi:10.3390/sports12080214_

Round 1

Reviewer 1 Report

Comments and Suggestions for Authors

Comments on the Quality of English Language

Author Response

We would like to thank the editors and reviewers for the kind consideration of our manuscript, and providing us with the very relevant comments.

Following are our responses to the comments from the reviewer:

Abstract

Lines 25-26: FMS assesses shoulder and lower body mobility; therefore, I find this correlation unnecessary to present especially in the abstract. The way it is presented is misleading as those parameters are assessed through FMS, so we expect them to show a correlation. Furthermore, shoulder and lower body mobility are not considered performance variables. It is not always necessary to show a significant correlation to publish research papers.

Response: Thank you for the suggestion. Change was made accordingly in manuscript We can write ‘Jump and speed performance score correlated well with ASLR, while the overall FMS score was not associated to any of the performance variable.’

Introduction

Lines 37-38: which tests are you referring to? There are several valid and reliable tests (field and laboratory tests) to assess the aforementioned parameters, therefore this sentence is not accurate.

Response: Thank you for the suggestion. The sentence has been deleted.

Lines 56-60: you are just repeating what you already said about FMS. Also, you need to support this statement as several very well-designed studies suggest that FMS does not relate in any aspect to athletic performance (Parchmann CJ, McBride JM. Relationship between functional movement screen and athletic performance. J Strength Cond Res. 2011, 25(12):3378-84. doi: 10.1519/JSC.0b013e318238e916).

I am not sure that the rationale of the study is clearly presented and supported. In the introduction, you focused a lot on the injury part (even on asymmetries), while none of the parameters that you assessed had anything to do with injuries or asymmetries. I was expecting to see at least a unilateral assessment of jumping performance. Single-leg jumps might have been more relevant to the asymmetries assessed through the FMS. I know the same tests were used in the study by Zhang and Colleagues (2022) but in their case, they didn’t emphasize the injury part so much in their introduction. I believe the introduction should be re-structured (you can use the following study as an example Zhang, J.; Lin, J.; Wei, H.; Liu, H. Relationships between Functional Movement Quality and Sprint and Jump Performance 325 in Female Youth Soccer Athletes of Team China. Children 2022, 9, 1312. https://doi.org/10.3390/children9091312).

Response: Thank you for the suggestion. The paragraph has been changed to focus on FMS and athletic performance.

The methodology and results sections are fine.

Response: Thank you.

Discussion

Lines 185-186: I am not sure how this statement is relevant to the purpose of this study. Did you assess the sensitivity of the test?

Response: We appreciate your observation. The statement refers to the findings of Zhang et al. [7], which revealed that an FMS score over the proposed threshold of 14.5 is indicative of strong sensitivity and a lower false-positive rate of FMS in female football players. While our study did not specifically evaluate the test's sensitivity, this remark offers context by connecting our findings with existing studies.

Line 194: why do you refer to your results as inconclusive? It is fine not to find a significant correlation.

Response: We thank the reviewer for the given suggestion. We have corrected it. We changed the term “inconclusive” to “ambiguous”.

Lines 197-198: which groups are you referring to?

Response: We added explanation “playing positions”.

Line 200: You need to specify on which tests your participants performed better compared to the Chinese athletes of Zhang et al. The fact that they performed better on the performance tests is expected but that doesn’t explain why there was no correlation between performance and FMS as indicated by Zhang et al.

Response: We thank the reviewer for his contributions to making the manuscript better. We added “and had better sprint and FMS score but lower jump performance”.

Lines 209-211: I am not sure how the results of Bennett support the importance of your assumption. I think you should either explain that further or rephrase as it doesn’t make sense.

Response: Thank you for the suggestion. The sentences have been rephrased as “Moreover, Bennett et al. [21] demonstrated the potential of FMS during athletic per-formance optimization since FMS can determine acceptable movement capabilities.”

Lines 237-238: this is an assumption which is not supported. You can keep this part if you have scientific support for it.

Response: Thank you for the suggestion. Change was made accordingly in manuscript. We added a reference (Kritz M, Cronin J, Hume P. The bodyweight squat: A movement screen for the squat pattern. Strength & Conditioning Journal. 2009 Feb 1;31(1):76-85.)

Lines 246-248: you haven’t indicated any correlations between FMS and performance parameters. This sentence is not accurate.

Response: Thank you for your feedback. We appreciate your observation regarding the correlations. As indicated in Table 3, we have demonstrated the correlations between FMS scores and various performance parameters. We hope this clarifies the presentation of our findings.

Your conclusions should concentrate more on your findings.

Response: Thank you for the suggestion. Change was made accordingly in manuscript.

Reviewer 2 Report

Comments and Suggestions for Authors

General comment: The authors investigated the correlations between functional movement quality (FMS) and speed, agility and jump performance in elite female youth football players.  The paper is of interest for practitioners dealing with young athletes as movement proficiency testing may provide useful information identifying probable functional deficits of the athletes that could lead to injuries or impaired performance.  The study is well-written and well-discussed. However, there are sections of the paper that need editing for clarity.

L 83: How was the number of participants determined? Please explain. Did you use a power test?

What was the maturation level of the adolescent athletes?

What was their training experience?

Did they have resistance training experience?

L 97-98: The tests took place after a 5-day national training camp. Did you check fatigue? Fatigue could have influenced the results of the study. If not, I suggest adding this to the limitations of your study.

L 125: Please provide validity and reliability scores for the certain photocells used in young athletes.

L 127-128: Why the photocell height was set at 120 cm? Please explain and provide relative reference.  

L 132: Was the agility Zig-Zag test time recorded with the same photocells? Please add relative information.

L 127, 136, 143, 153:  Please provide the reliability scores of the trials performed in this study for sprint, Zig-Zag test, T-test and CMJ.

L 147: Why did you use CMJ free hands? What about variability of performance because of arm swing? Furthermore, the reference used reports CMJ with hands in akimbo position, please replace it with one using CMJ with arm swing.

L 170: There are no morphological parameters in Table 3. Please delete relative information.

L 178 Table 3 Notes: BMI: Body mass index; PBF: Percentage of body fat; FFM: Fat free mass; are not presented in Table 3. Please delete them.

L 187-188: You report that FMS score had a very weak positive influence on performance indices. Was that statistically significant? Your results showed no overall correlation between FMS score and performance. Please rephrase.

L 195-196: Please first estimate participants’ maturity offset and then discuss accordingly.

L 226-239: Please provide references for the information provided in this paragraph.

L 274: Please add a paragraph discussing the limitations of the study.

Author Response

We would like to thank the editors and reviewers for the kind consideration of our manuscript, and providing us with the very relevant comments.

Following are our responses to the comments from the reviewer:

L 83: How was the number of participants determined? Please explain. Did you use a power test?

What was the maturation level of the adolescent athletes?

What was their training experience?

Did they have resistance training experience?

Response: We thank the reviewer for his contributions to making the manuscript better. Here are the details regarding to the questions:

  • All of the participants were members of the Bosnia and Herzegovina national U16 team. The national coach selected all team members, and they were subsequently tested. Therefore, the total number of competitors in the national selection was the determining factor for the number of participants.
  • We did not perform a formal power analysis for this study. The sample size was inherently restricted by the availability of the national team members, which defined the participant pool.
  • The maturation levels of the adolescent athletes were not specifically known to us. Nevertheless, the national U16 team selected all participants, who were of similar age, indicating a certain level of physical and developmental maturity that is appropriate for high-level competition.
  • There was a minimum of five years of training experience among all participants. We added a sentence in the text.
  • Although we did not have access to comprehensive records of their resistance training experience, it is reasonable to assume that they were familiar with and exposed to resistance training as part of their overall athletic development as members of a top-tier national selection.

L 97-98: The tests took place after a 5-day national training camp. Did you check fatigue? Fatigue could have influenced the results of the study. If not, I suggest adding this to the limitations of your study.

Response: We appreciate your observations regarding the potential impact of fatigue on the results of our study. The tests were taken following a 5-day nationwide training camp. No vigorous workouts were conducted throughout this period. The training camp mostly emphasized tactical and technical skills, while avoiding high-intensity or demanding physical activities for the players.

L 125: Please provide validity and reliability scores for the certain photocells used in young athletes.

Response: Thank you for the suggestion. Change was made accordingly in manuscript.

L 127-128: Why the photocell height was set at 120 cm? Please explain and provide relative reference.  

Response: Thank you for the suggestion. Change was made accordingly in manuscript.

L 132: Was the agility Zig-Zag test time recorded with the same photocells? Please add relative information.

Response: Thank you. We added “using same photocells (Microgate, Bolzano, Italy)”.

L 127, 136, 143, 153:  Please provide the reliability scores of the trials performed in this study for sprint, Zig-Zag test, T-test and CMJ.

Response: Thank you for the suggestion. Change was made accordingly in manuscript, we have added ICC for every test and provide some references.

L 147: Why did you use CMJ free hands? What about variability of performance because of arm swing? Furthermore, the reference used reports CMJ with hands in akimbo position, please replace it with one using CMJ with arm swing.

Response: Thank you for the suggestion. Change was made accordingly in manuscript, we added references. Allowing the use of arms during the CMJ exam mirrors football players' natural movement patterns. In a real-game scenario, players frequently utilize their arms for balance and to generate extra force when jumping. Testing with free hands allows for a more accurate assessment of their actual jumping performance on the field.

L 170: There are no morphological parameters in Table 3. Please delete relative information.

Response: Thank you for the suggestion. Change was made accordingly in manuscript.

L 178 Table 3 Notes: BMI: Body mass index; PBF: Percentage of body fat; FFM: Fat free mass; are not presented in Table 3. Please delete them.

Response: Thank you for the suggestion. Change was made accordingly in manuscript.

L 187-188: You report that FMS score had a very weak positive influence on performance indices. Was that statistically significant? Your results showed no overall correlation between FMS score and performance. Please rephrase.

Response: Thank you for the suggestion. Change was made accordingly in manuscript.

L 195-196: Please first estimate participants’ maturity offset and then discuss accordingly.

Response: Thank you for the suggestion. Change was made accordingly in manuscript.

L 226-239: Please provide references for the information provided in this paragraph.

Response: Thank you for the suggestion. Change was made accordingly in manuscript. We added references (Kritz M, Cronin J, Hume P. The bodyweight squat: A movement screen for the squat pattern. Strength & Conditioning Journal. 2009 Feb 1;31(1):76-85.; Mašić S, ÄŒaušević D, ÄŒović N, Spicer S, Doder I. The benefits of static stretching on health: a systematic review. Journal of Kinesiology and Exercise Sciences. 2024 Jan 12:1-7.)

L 274: Please add a paragraph discussing the limitations of the study.

Response: Thank you for the suggestion. Change was made accordingly in manuscript.

Round 2

Reviewer 1 Report

Comments and Suggestions for Authors

I am happy with the changes and revisions. I have no further comments or recommendations. 

Comments on the Quality of English Language

Some minor issues may be addressed with a native English speaker proofreading the work. 

Reviewer 2 Report

Comments and Suggestions for Authors

I appreciate the authors' thorough response to the raised issues.. The changes implemented address the concerns raised during the initial review process, leading to a notable improvement in the manuscript. The authors' efforts in revising the text have strengthened the impact of the study.